# URLs Help, Topics Guide:
# Understanding Metadata Utility in LLM Training

**Dongyang Fan   Vinko Sabolčec   Martin Jaggi**
EPFL, Switzerland
`firstname.lastname@epfl.ch`

## Abstract

Large Language Models (LLMs) are commonly pretrained on vast corpora of text without utilizing contextual metadata such as source, quality, or topic, leading to a context-free learning paradigm. While recent studies suggest that adding metadata like URL information as context (i.e., auxiliary inputs not used in the loss calculation) can improve training efficiency and downstream performance, they offer limited understanding of which types of metadata are truly effective and under what conditions. In this work, we conduct a systematic evaluation and find that not all metadata types contribute equally. Only URL context speeds up training, whereas quality scores and topic/format domain information offer no clear benefit. Furthermore, the improved downstream performances of URL conditioning emerge only when longer prompts are used at inference time. In addition, we demonstrate that context-aware pretraining enables more controllable generation than context-free pretraining, in a classifier-free guidance fashion. Although topic and format metadata do not accelerate training, they are effective for steering outputs, offering human-interpretable control over generation.

## 1   Introduction

Large Language Models (LLMs) have become increasingly integrated into a wide range of real-world applications, prompting both researchers and practitioners to explore optimal strategies for their development. A critical component of this process is the pretraining phase, which serves as the foundational stage in which models acquire general-purpose linguistic and world knowledge from vast corpora of text. Even with a simple next-token prediction loss, the scale of data and model size enables emergent generalization of knowledge [1].

Conventional LLM pretraining is typically *context-free*, in the sense that training data consists solely of raw textual inputs, without the inclusion of any associated metadata or external contextual signals. Meta information, such as document source, author identity, timestamps, or topic tags—although potentially valuable—are systematically discarded during preprocessing. As a result, the model learns to predict the next token based on the intrinsic structure and statistical patterns present within the text itself, without leveraging any auxiliary semantic or structural cues. A natural research question arises: Can the pretraining be improved if contextual information is included?

While prior studies [2, 3] suggest that incorporating URLs as context (auxiliary inputs not used in the loss calculation) during training can accelerate pretraining and enhance downstream performance, recent findings by Higuchi et al. [4] caution that this is not always the case. Existing work lacks a comprehensive understanding of *which types of metadata are most impactful and under what conditions they offer the greatest benefit*. Our study aims to fill this gap, providing empirical evidence based on real-world experiments.

39th Conference on Neural Information Processing Systems (NeurIPS 2025).

Fundamentally, context-aware pretraining endowed LMs with the ability to perform conditional autoregressive generation. This conditional generation capability, developed through exposure to metadata, can potentially be leveraged to steer the model's behavior using a classifier-free guidance approach [5], i.e., the distinction between context-conditioned and context-free generation can be further amplified to highlight the influence of contextual information. We aim to investigate *whether context-aware pretraining can be utilized for more controllable generation than context-free pretraining.*

In response to the previously raised questions, our findings and contributions can be summarized as follows:

- We conduct a comprehensive study on context-conditioned LLM pretraining, highlighting its potential benefits and limitations at both pretraining and inference stages (Section 3).
- We examine different types of metadata and find that only URL as context can speed up pretraining (Section 4.1.1). When it comes to downstream evaluation, URL-conditioned pretraining only enhances the performance with longer prompts (Section 4.1.2).
- We demonstrate that the metadata-conditioned pretrained model is more steerable than the standard pretrained model. While topic and format domain information does not accelerate pretraining, it offers effective and human-interpretable steering (Section 4.2).

## 2 Related Work

**Metadata for LLM pretraining.** Allen-Zhu and Li [2] demonstrate through controlled synthetic experiments that adding a special token to high-credibility data can be beneficial. They further hypothesize that appending domain names (e.g., wikipedia.org) to each document in the pretraining corpus would yield similar gains. This hypothesis is experimentally validated by Gao et al. [3], who report a 33% speedup in pretraining simply by prepending URL domains. However, to ensure the model remains effective on standard, domain-free text, they introduce a "cool-down" phase using non-contextual data. Following this, Zhu et al. [6] formally proves that context-enhanced learning can be exponentially more sample-efficient when the model is capable of in-context learning, that is, it can effectively utilize contextual information presented during inference to perform tasks. More recently, Higuchi et al. [4] caution that metadata conditioning is not universally effective. Buidling up on the experimental setup of [2], they found that metadata conditioning only helps when the input prompt is sufficiently long to reveal latent semantics. With shorter prompts, it can even degrade performance. Most prior studies conduct experiments solely on synthetic datasets, where the metadata consists of the datasets' production rules. However, in real-world scenarios, metadata cannot fully capture the complexity of natural language. Therefore, we aim to explore the extent to which such metadata can actually be helpful.

**Guidance in LLMs.** In LLMs, next-token generation is essentially a sampling process from a probability distribution over the entire vocabulary. This makes it possible to steer the generation by modifying the token probability distribution. Liu et al. [7] guide the generation of a base LLM by leveraging the difference between the outputs of an expert model and an anti-expert model. Li et al. [8] adjust the sampling distribution by down-weighting tokens favored by an amateur model. Sanchez et al. [9] introduced the idea of classifier-free guidance (CFG)—originally used in text-to-image generation—to text-only generation. Building on this, He et al. [10] applied CFG to steer LLMs for personalization purposes, showing potential for controllable personalization. However, all of these approaches rely on context-free pretrained LLMs. In this work, we investigate can context-conditioned pretrained LLMs enable more effective context steering.

## 3 Method

### 3.1 Context-Conditioned Pretraining

To properly organize the contextual information, we introduce two new tokens: `<|context_begin|>` and `<|context_end|>`[1]. The meta information is inserted between the two tokens, and prepended

---

[1]for illustration purposes, we use `<boc>` and `<eoc>` to represent these two tokens

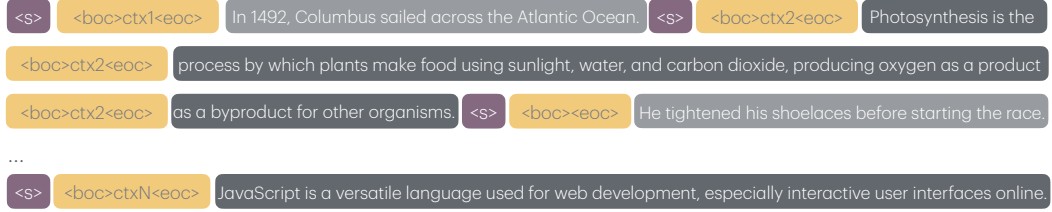

Figure 1: An example of our context-aware tokenization. Each document begins with a default beginning-of-sequence (``) token. For each sequence, a context segment wrapped in beginning-of-context (`<boc>`) and end-of-context (`<eoc>`) is inserted after `` and before the main text. If a document is too long and split into multiple sequences, the context is prepended to each one. Although the context is added to every sequence, it may be empty. In 90% of the corpus, we include a non-empty context; in the remaining 10%, the context is left empty. Contexts can be URL, quality score, or topic/format domains depending on user choices.

before the document contents. During training time, we mask out the loss calculation for the prepended contexts, so the loss is solely calculated over the standard texts, making it comparable to standard context-free pretraining. To avoid confusion, we use *context* to refer to the auxiliary meta information not used in the loss calculation during training time, and use *prompts* for the texts given to the pretrained model for generation.

Context is introduced at the sequence level and is specific to each document. To ensure the model remains capable of handling context-free text at inference time, we uniformly interleave context-prepended and context-free documents during pretraining, using a 90%:10% ratio. The context enhanced corpus is illustrated in Figure 1, where one document can be split into several sequences and contexts are prepended to each of them. Unlike the MeCo approach [3], which applies 90% context-prepended training followed by a 10% context-free cooldown phase, our uniform mixture strategy allows any intermediate checkpoint to be immediately used for context-free inference.

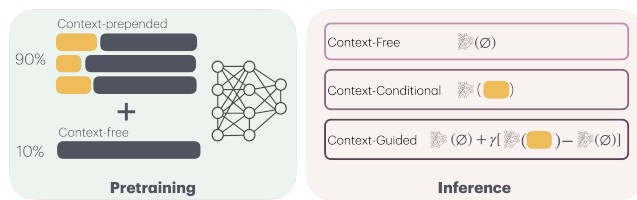

Figure 2: Diagram of our two-stage investigation. During pretraining time, we feed a uniform mixture of 90% context-prepended texts and 10% context-free standard texts into the model. During inference time, we compare three different generation sampling methods.

## 3.2 Context-Aware Generation

At inference time, leveraging the conditioning capabilities learned during pretraining, we can guide the LLM to generate text based on a given context. Next-token prediction involves sampling from a probability distribution ($\mathcal{P}^{\mathrm{LLM}}$) over the entire vocabulary. To incorporate contextual signals, we modify this distribution accordingly. We define the three following sampling methods for generation:

1) Context-Free sampling: this is the standard sampling given by an LLM, conditioning on empty contexts.
$$x^t \sim \mathcal{P}^{\mathrm{LLM}}(x|\boldsymbol{x}^{1:t-1}, \emptyset) \qquad \text{(ctx-free)}$$

2) Context-Conditioned sampling: for each generation, we prepend `<boc>ctx<eoc>` to the prompt.
$$x^t \sim \mathcal{P}^{\mathrm{LLM}}(x|\boldsymbol{x}^{1:t-1}, \mathrm{ctx}) \qquad \text{(ctx-conditioned)}$$

3) Context-Guided sampling: following the classifier-free guidance in diffusion models [5], we amplify the impact of the context with the following sampling probability:
$$x^t \sim \mathcal{P}^{\mathrm{LLM}}(x|\boldsymbol{x}^{1:t-1}, \emptyset) \cdot \left( \frac{\mathcal{P}^{\mathrm{LLM}}(x|\boldsymbol{x}^{1:t-1}, \mathrm{ctx})}{\mathcal{P}^{\mathrm{LLM}}(x|\boldsymbol{x}^{1:t-1}, \emptyset)} \right)^{\gamma} \qquad \text{(ctx-guided)}$$

During inference, in practice, we modify the generation in the logit ($\Pi$) level, that is

$$x^t \sim \Pi^{\text{LLM}}(x|\boldsymbol{x}^{1:t-1}, \emptyset) + \gamma \left( \Pi^{\text{LLM}}(x|\boldsymbol{x}^{1:t-1}, \text{ctx}) - \Pi^{\text{LLM}}(x|\boldsymbol{x}^{1:t-1}, \emptyset) \right) \qquad (1)$$

When $\gamma = 0$, this coincides with context-free generation and when $\gamma = 1$, this coincides with context-conditioned generation. With $\gamma > 1$, we amplify the guidance of the context.

It is worth noting that the softmax output after context-guidance is no longer a faithful[2] probability distribution, while context-free and context-conditioned generation still follow a faithful probability distribution.

## 4 Experiments

**Model.** We adopt the Llama model architecture [11] with 16 layers, a hidden size of 2048, a sequence length of 4096, and a batch size of 504 (resulting in 2.06 million tokens). The model has 1.5 billion parameters. We follow the Cosine learning schedule, applying 2000 warmup steps. AdamW optimizer is used with regularization strength 0.1 [12]. A max learning rate of 3e-4 is applied and cools down to learning rate 3e-5 at the end of training. To train the models, we use the Megatron-LM framework [13].

**Dataset.** We use FineWeb-Edu dataset [14], which is a high-quality English-only dataset. The dataset has a lot of meta data available, such as URL source, quality score and token counts per document. We sub-sample FineWeb-Edu and tokenize it using Nemo tokenizer[3], with two additionally introduced tokens. Throughout training, we randomly sample 100B tokens.

**Evaluation Benchmarks**. As standard practice, we evaluate models on general knowledge understanding using LM-Eval-Harness developed by Gao et al. [15]. The benchmarks used are Arc-Easy [16], Arc-Challenge [16], CommonSense QA (CSQA, 17), MMLU [18], PIQA [19], Social IQA (SIQA, 20), HellaSwag (HS, 21), Lambada (LBD, 22) and Winogrande (WG, 23).

We compare the following runs to gain a full insight over how can contexts potentially enhance LLM pretraining.

1. Baselines
   - *Standard Pretraining*: Each document is simply prepended with a begin-of-sequence token .
   - *Positional Token Ablation*: Standard pretraining where each sequence is prepended with <boc><eoc> tokens. This setup tests whether introducing a positional token at the start of each sequence alone has a beneficial effect.
   - *MeCo*: A two-phase approach proposed by Gao et al. [3]—90% of training is URL-conditioned, followed by a 10% cooldown phase using standard (unconditioned) data.

2. Types of Metadata
   - *Full URL (URL)*: The original source from which the document was crawled.
   - *Quality Score (QS)*: Provided by the FineWeb-Edu dataset, these scores are generated by a classifier trained on 450,000 web samples, each annotated by LLaMA 3 with a score from 0 (not educational) to 5 (highly educational), reflecting the sample's educational value.
   - *Domain Information (DI)*: Each document is labeled with topic and format domains derived from WebOrganizer [24], where topic and format domains are returned by pretrained classifiers. There are 24 categories per taxonomy. In total, we have 576 different Domain Information types.

3. Combinations of Metadata
   - *Full URL + Quality Score*: Combines the document's source with its educational quality.
   - *Full URL + Domain Information*: Combines the source with topic and format domain labels.

---

[2]By faithful, we mean a probability distribution that can be trusted as a true representation of uncertainty, without distortion, manipulation, or miscalibration.

[3]https://mistral.ai/news/mistral-nemo

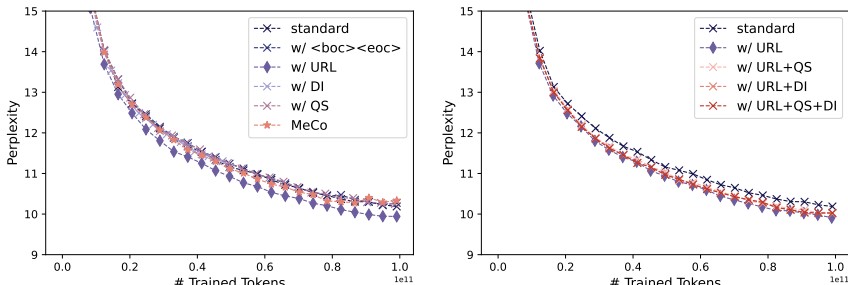

Figure 3: Training perplexity versus the amount of consumed tokens. Prepending the URL leads to a faster decrease in perplexity.

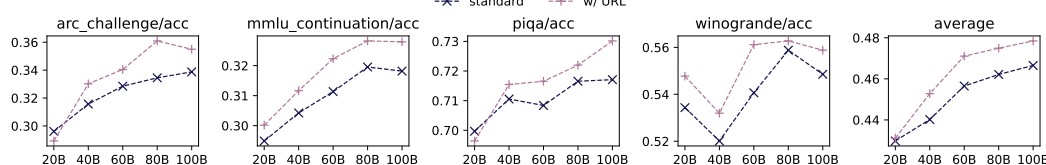

Figure 4: URL-conditioned pretraining achieves the same downstream evaluation performances of 100B-token standard pretraining with only 60B tokens. The same plots with respect to all tasks are provided in Figure 7.

- *Full URL + Quality Score + Domain Information*: Incorporates all available metadata to condition the model on source, quality, and content structure.

## 4.1 Not All Metadata Conditioning Speed-up Pretraining

### 4.1.1 Training Speed-up with URL Conditioning

We begin by investigating whether metadata can facilitate training, specifically, whether pretraining can be accelerated when additional contextual information aids knowledge acquisition. The left panel of Figure 3 presents the training perplexity across runs with different metadata types. The two additional tokens introduced have minimal effect on learning dynamics, indicating that any observed differences stem from the contextual content itself. Among the three types of context evaluated, only the URL consistently results in lower perplexity from the very beginning of training.

**Is the improved performance stemming from longer contexts?** Given that URLs are substantially longer than the other two metadata types, one might wonder whether their effectiveness stems merely from increased context length. To examine this, we combined the other metadata types with URLs to see if additional context would further improve training. From the right panel of Figure 3, the answer is No. This indicates that it is not the increased context length that speeds up the training, but the context content itself.

**Does the acceleration translate into downstream performance as well?** With five-shot evaluation, we see better performance for the URL-conditioned model in every task. For the 9-task average, the URL-conditioned model can match the performance of the standard pretrained model on 100B tokens with only 60B tokens, achieving a 40% significant acceleration, as shown in Figure 4. The acceleration in terms of downstream performance is greater than training loss.

**Why only URL helps?** We have already controlled for the confounding factor — the length of metadata. However, it remains unclear why only the URL contributes meaningfully. To gain insight into this, we visualized the attention patterns of the document texts toward different parts of their corresponding contexts in Figure 5. Except for the URL-conditioned model, the other models fail to allocate substantial attention to the informative parts of the contexts in the early layers. We *hypothesize* that the early-layer attention to meaningful URL components may play a key role in the effectiveness of URL prepending, even if later layers focus on less informative tokens. Moreover, it is important to recognize that URLs inherently encode both domain information and document quality,

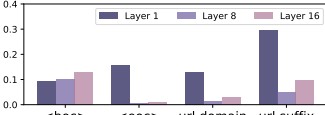 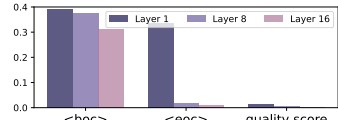 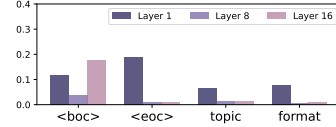

Figure 5: Average attention to different parts of different prepended contexts. Take `https://en.wikipedia.org/wiki/Metadata#Standards` as an example. `en.wikipedia.org` is the URL domain and `/wiki/Metadata#Standards` is the URL suffix. More details in Appendix C.1.2.

which may differ from human-assigned DI and QS labels. This distinction likely also contributes to the superior performance of the URL feature.

> **Takeaway 1.** Only URL conditioning has a positive effect in speeding up pretraining; conditioning on quality scores or topic/format domains yields no noticeable change.

### 4.1.2 Downstream Performance Gains from Longer Prompts

**When does URL conditioning work?** To better understand the impact of context conditioning, we perform a systematic evaluation across downstream tasks by comparing zero-shot and five-shot performance. We notice a very interesting phenomenon: *The URL-conditioned pretrained model shows increased downstream evaluation performance with five-shot evaluation, but not zero-shot.* For the rest, we do not see substantial improvement over standard pretraining. Our results confirm recent findings from Higuchi et al. [4], where longer prompts are needed to infer latent semantics and thus to help evaluation.

**Conflicting signals of different metadata types.** Interestingly, the benefits of URL conditioning are negated when additional metadata is introduced as context. For example, combining quality scores with URL conditioning appears to introduce conflicting signals, leading to downstream performance that is even lower than that of the standard pretrained model. This suggests that the latent clusters inferred by the language model from the URLs may differ from the predefined clusters we assigned. Notably, we did not manage to reproduce the results of the MeCo baseline. Possible reasons could be different datasets and models, and that LM-Eval-Harness [15] is used as our evaluation framework is used in our experiments, while OLMES [25] was used in the MeCo paper. Nonetheless, we still observe the speeding-up effect of URL prepending.

Table 1: 0shot evaluation results. When evaluating, context-free generation is used, i.e., only `<boc><eoc>` tokens are prepended and the context string is empty.

|  | **Arc-C** | **Arc-E** | **CSQA** | **MMLU** | **PIQA** | **SIQA** | **HS** | **LBD** | **WG** | **Avg** |
|---|---|---|---|---|---|---|---|---|---|---|
| standard | 32.9 | 68.9 | 42.3 | 31.2 | 71.3 | 40.6 | 42.6 | 33.0 | 57.1 | 46.7 |
| + `<boc><eoc>` | 33.1 | 68.4 | 41.8 | 32.0 | 72.1 | 40.2 | 42.4 | 33.4 | 56.6 | 46.7 |
| + `<boc>` URL `<eoc>` | 33.5 | 69.8 | 41.6 | 32.0 | 73.4 | 41.3 | 42.8 | 33.2 | 54.4 | **46.9** |
| + `<boc>` QS `<eoc>` | 30.9 | 68.4 | 38.3 | 30.9 | 72.4 | 41.0 | 42.0 | 34.4 | 53.5 | 45.8 |
| + `<boc>` DI `<eoc>` | 32.1 | 68.4 | 41.9 | 31.8 | 71.3 | 41.1 | 42.1 | 34.2 | 53.4 | 46.3 |
| + `<boc>` URL, QS `<eoc>` | 31.2 | 68.7 | 38.2 | 31.7 | 71.3 | 40.1 | 42.2 | 34.6 | 54.2 | 45.8 |
| + `<boc>` URL, QI `<eoc>` | 31.2 | 68.1 | 37.8 | 31.8 | 71.9 | 40.6 | 42.0 | 33.7 | 56.5 | 46.0 |
| + `<boc>` URL, QS, DI `<eoc>` | 32.4 | 69.4 | 40.8 | 31.8 | 71.8 | 39.8 | 42.0 | 33.0 | 54.9 | 46.2 |
| MeCo | 33.0 | 68.9 | 39.7 | 31.8 | 71.3 | 40.1 | 41.9 | 33.0 | 55.6 | 46.2 |

> **Takeaway 2.** URL-conditioned pretraining only benefits evaluation for longer prompts, for example, 5-shot. There is no noticeable distinction for 0-shot evaluation.

## 4.2 Context-Aware Generation

### 4.2.1 Context-Conditioned Generation

The results from Table 2 are from context-free generation using a context-conditioned model. We now test whether the performance can further be improved if correct contexts are given at test time.

Table 2: 5shot evaluation results. When evaluating, context-free generation is used, i.e., only `<boc><eoc>` tokens are prepended and the context string is empty.

| | Arc-C | Arc-E | CSQA | MMLU | PIQA | SIQA | HS | LBD | WG | Avg |
|---|---|---|---|---|---|---|---|---|---|---|
| standard | 33.9 | 68.6 | 45.1 | 31.8 | 71.7 | 41.5 | 41.9 | 30.6 | 54.9 | 46.7 |
| + `<boc><eoc>` | 34.0 | 69.2 | 43.5 | 31.9 | 71.9 | 41.5 | 42.5 | 30.8 | 55.2 | 46.7 |
| + `<boc>` URL `<eoc>` | 35.5 | 71.7 | 46.8 | 32.8 | 73.0 | 41.7 | 42.5 | 30.8 | 55.9 | **47.8** |
| + `<boc>` QS `<eoc>` | 32.7 | 69.6 | 41.0 | 31.8 | 72.1 | 41.6 | 43.0 | 32.4 | 55.2 | 46.6 |
| + `<boc>` DI `<eoc>` | 34.0 | 69.8 | 42.6 | 32.1 | 71.9 | 40.3 | 42.0 | 31.8 | 55.5 | 46.7 |
| + `<boc>` URL, QS`<eoc>` | 32.5 | 67.8 | 42.8 | 31.6 | 71.4 | 41.7 | 41.8 | 32.0 | 53.0 | 46.1 |
| + `<boc>` URL, DI `<eoc>` | 33.9 | 67.4 | 42.2 | 31.4 | 71.8 | 41.5 | 42.3 | 32.9 | 57.0 | 46.7 |
| + `<boc>` URL, QS, DI `<eoc>` | 32.9 | 68.6 | 44.6 | 31.9 | 71.6 | 39.8 | 41.9 | 31.8 | 51.0 | 46.0 |
| MeCo | 32.6 | 69.7 | 46.3 | 32.5 | 72.0 | 39.9 | 41.6 | 31.3 | 54.0 | 46.7 |

For each task, we conditioned the generation on an contexts that match the task content. We manually select these contexts. For Quality Score context, we simply prepend `Quality Score: 5` in the inference time. The contexts we used are presented in Table 10 in the Appendix. Conditioning on provided contexts, we observed increased downstream performance, especially in URL- and DI-conditioned pretrained models.

Table 3: Enhanced downstream task performance with context-conditioned generation compared to context-free generation. Three different contexts are tested here, that is, URL, Domain Information (DI) and Quality Score (QS).

| Model | Context | Arc-C | Arc-E | CSQA | MMLU | PIQA | SIQA | HS | LBD | WG | Avg |
|---|---|---|---|---|---|---|---|---|---|---|---|
| URL-conditioned | w/o context | 35.5 | 71.7 | 46.8 | 32.8 | 73.0 | 41.7 | 42.5 | 30.8 | 55.9 | 47.8 |
| | w/ URL context | 35.5 | 72.2 | 47.7 | 32.9 | 72.3 | 43.0 | 43.0 | 32.5 | 56.0 | 48.3 ↑ |
| DI-conditioned | w/o context | 34.0 | 69.8 | 42.6 | 32.1 | 71.9 | 40.3 | 42.0 | 31.8 | 55.5 | 46.7 |
| | w/ DI context | 33.9 | 70.9 | 46.0 | 32.3 | 72.4 | 41.7 | 42.2 | 33.2 | 55.1 | 47.5 ↑ |
| QS-conditioned | w/o context | 32.7 | 69.6 | 41.0 | 31.8 | 72.1 | 41.6 | 43.0 | 32.4 | 55.2 | 46.6 |
| | w/ QS context | 32.0 | 70.0 | 44.3 | 32.1 | 72.7 | 42.4 | 42.0 | 32.9 | 54.4 | 47.0 ↑ |

#### 4.2.2 Context-Guided Generation

Can the impact of context be further amplified? Accuracy calculation in Table 3 relies on log-likelihood that requires probability distributions. Classifier-free guidance modifies the output logits before softmax to steer generation. The logits have been manually shifted, which makes the resulting probability distribution not grounded in the model's knowledge or data, thus breaking the probabilistic interpretation. Likelihood-base evaluation does not work. We turn to generation-based evaluation instead. Given the limited capacity of our 1.5B base models, we cannot evaluate them meaningfully using free-form generation for the QA tasks. As a result, we assess their ability to generate coherent continuations when provided with a prompt.

We gather prompts in health and history domains, and let our pretrained models to finish the prompts. We let `gpt-4o` evaluate over 4 dimensions regarding the continuation, that are, coherence, correctness, reasonableness, and relevance. The scores are between 1 and 10, with higher score being more desirable. With context-guided generation, we apply the steering factor $\gamma = 1.5$ if not mentioned, from a hyperparameter grid search in Figure 6. Over 50 prompts, we have the `gpt-4o` evaluation presented in Table 4, where we compare the three different generation methods using both our context-conditioned pretrained model and standard context-free pretrained model. Given that LLM judges may hallucinate, we run a 3-seed evaluation and report the mean and standard deviation. Com-

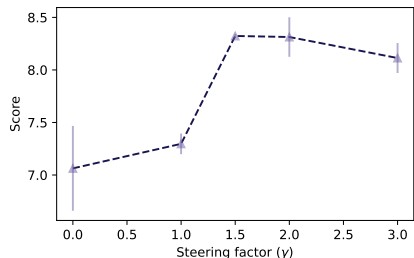

Figure 6: LLM judge scores (with standard deviation) vs. different values of steering factor $\gamma$

pared to the standard pretrained model, we observe that context-conditioned pretrained model is more steerable, as showcased by improved performance with context-guided generation.

**Which metadata type is efficient in guidance?** Among the three different context types, we notice that even though topic and format domains are not effective in speeding up training, they offer effective steering possibilities. We provide an example of different continuations from the three

different sampling methods in Table 5. Conditioning on `Topic: Health, Format: Article`, while context-free generation offers very generic answers, context-condiitoned generation is able to offer more health-relevant technical answers. Context-guided generation is the most comprehensive and informative among the three, indicating the successful guidance of the context.

Table 4: Average LLM-as-a-judge evaluation scores (mean $\pm$ std) on the different prompt completions on 50 prompts on health & medical and history topics respectively (answer length: 64 tokens). Model type denotes the type of checkpoints used, with "conditioned" being our context-conditioned pretrained checkpoint and "standard" being the standard context-free pretrained checkpoint. For each of the different contexts, we use different context-pretrained models.

| topic | context | model type | ctx-free | ctx-conditioned | ctx-guided |
|---|---|---|---|---|---|
| health & medical | *URL: https://medicalxpress.com/* | *Conditioned* | $8.19 \pm 0.14$ | $8.14 \pm 0.20$ | $8.70 \pm 0.14$ |
| | | *Standard* | $7.98 \pm 0.34$ | $7.79 \pm 0.10$ | $8.12 \pm 0.26$ |
| | *Topic: Health, Format: Knowledge Article* | *Conditioned* | $8.01 \pm 0.16$ | $8.26 \pm 0.14$ | $\mathbf{9.07 \pm 0.05}$ |
| | | *Standard* | $7.75 \pm 0.03$ | $7.79 \pm 0.08$ | $7.66 \pm 0.24$ |
| | *Quality Score: 5* | *Conditioned* | $8.20 \pm 0.19$ | $8.20 \pm 0.20$ | $8.69 \pm 0.14$ |
| | | *Standard* | $8.37 \pm 0.20$ | $8.22 \pm 0.23$ | $8.09 \pm 0.21$ |
| history | *URL: https://britannica.com/* | *Conditioned* | $5.70 \pm 0.15$ | $5.80 \pm 0.16$ | $6.73 \pm 0.43$ |
| | | *Standard* | $4.88 \pm 0.15$ | $5.60 \pm 0.14$ | $6.45 \pm 0.01$ |
| | *Topic: History, Format: Knowledge Article* | *Conditioned* | $5.35 \pm 0.24$ | $5.23 \pm 0.26$ | $\mathbf{7.15 \pm 0.26}$ |
| | | *Standard* | $5.19 \pm 0.31$ | $5.95 \pm 0.11$ | $6.04 \pm 0.31$ |
| | *Quality Score: 5* | *Conditioned* | $5.36 \pm 0.20$ | $5.67 \pm 0.16$ | $6.45 \pm 0.34$ |
| | | *Standard* | $5.52 \pm 0.19$ | $5.68 \pm 0.19$ | $7.04 \pm 0.09$ |

Table 5: Example of different continuations (answer length: 64 tokens) given a prompt and their corresponding ratings. The Domain Information context provided is `Topic: Health, Format: Knowledge Article`. Continuations are generated from our DI-conditioned pretrained checkpoints. (Caution of Hallucination!)

| | *Given prompt*: Common causes of balance problems in older people include | Rating/Short Rationale |
|---|---|---|
| standard | falls and underlying conditions. These can range from minor health issues to more serious problems. You can get balance problems from several causes, including poor nutrition, aging, and underlying health conditions like high blood pressure or thyroid problems. It's also possible to suffer from balance problems suddenly. | 5.5. Clear but general and slightly repetitive |
| context-conditioned | medications, ear problems, and spinal and neck problems. Vestibular rehabilitation physical therapy, which uses special exercises to enhance brain signals that help maintain balance, can be used to treat some types of balance disorders. In most cases, medical attention is necessary to treat this type of dizziness or imbalance. | 7.0. More technical, introduces helpful treatment concepts |
| context-guided | ageing-related changes in the inner ear, neurological problems, and problems with medication. Other reasons for balance problems include head injury, Parkinson's disease, multiple sclerosis, and stroke. Vitamin B12 deficiency can cause balance problems. | 9.0. Most comprehensive and informative |

**Controllable guidance level.** We further demonstrate various steering possibilities enabled by our method. For example, it is possible to guide the generation toward a desired topic by increasing the guidance factor value ($\gamma$). When conditioning on `Topic: Fashion & Beauty, Format: Personal Blog` and setting $\gamma > 1$, LLMs sometimes produce humorous or absurd outputs in Figure 6, such as stating that Julius Caesar was assassinated because "of his taste in makeup" or "he wore too much jewelry." While this clearly highlights the model's tendency to hallucinate, it also underscores the strength of the steering mechanism. With appropriate contextual grounding, such guidance could potentially yield coherent and relevant outputs in desired domains.

**Controllable style and content.** Moreover, when provided with different contextual signals, our models can adapt the continuation in ways that align with the intended framing or source. As shown

Table 6: Example of different context-guided continuations given a prompt and a steering factor ($\gamma$). The Domain Information context is `Topic: Fashion & Beauty, Format: Personal Blog`. Continuations are generated from our pretrained DI-conditioned checkpoints. (Caution of Hallucination!)

| Steering factor ($\gamma$) | *Given prompt*: Julius Caesar was assassinated because |
|---|---|
| -2 | he'd killed too many people and he'd become too dangerous. He was assassinated by senators and officials who were on the wrong side of the Senate. Caesar feared they might take power away from him and start an anarchy. |
| 0 | he was trying to become king of the Roman Empire. He was forced to flee to Greece to escape the Roman legions. This was a major mistake, because he was already an experienced general and knew that he could not defeat them in the long run. |
| 1 | his assassins feared he would be a threat to their power. At the time, the average Roman man's wardrobe consisted of a tunic and a short-sleeved shirt. But as soon as Julius Caesar was assassinated, Rome's dress code became stricter. |
| 2 | of his taste in makeup. Back then, makeup was not only for looks. It also had medicinal purposes. The Romans used makeup to soothe and protect the skin from the sun. It also had medicinal purposes. They believed it could heal certain skin conditions and infections. |
| 4 | he wore too much jewelry. Napoleon was so inspired by Alexander the Great that jewelry became a sign of royalty and he wore very little other than crowns, earrings, necklaces and bracelets. Kings, Queens, royalty and those with influence wore so much more than is necessary. |

in Table 7, the open-ended prompt "Paris is known for" leads to varied continuations depending on the guiding URL: `medicalxpress.com` prompts a medically-oriented response, `nytimes.com` results in a factual and formal description by listing facts, and `reddit.com` encourages a casual, entertainment-focused tone. A similar result with Domain Information guidance is presented in Table 8 and Table 9 in the Appendix, where we observe effective guidance in both topic and format domains. These results demonstrate that contextual steering can shape both the content and style of the generated text in meaningful ways.

> **Takeaway 3.** With context-conditioned pretraining, a larger improvement is observed in context-guided generation compared to context-conditioned generation.

> **Takeaway 4.** Although domain information does not accelerate pretraining, it provides more interpretable and efficient guidance during inference.

## 5 Conclusion

In this work, we present a comprehensive investigation into context-conditioned pretraining for LLMs, focusing on the impact of different metadata types. Our findings reveal that not all metadata types contribute equally to model efficiency or downstream performance. Specifically, URL metadata stands out as the only form of context that significantly accelerates training and improves performance—particularly when longer prompts are used at inference time. In contrast, quality scores and topic/format domain information do not yield measurable benefits during pretraining.

Nevertheless, we demonstrate that these latter forms of metadata have strong value during inference, enabling more controllable and interpretable text generation through classifier-free guidance mechanisms. This dual insight, that certain metadata can optimize training efficiency while others enhance controllability, opens promising directions for more flexible and human-aligned language model applications.

Table 7: Example of different continuations given a prompt conditioned on different URLs. The continuations are generated in a context-guided fashion using our URL-conditioned pretrained checkpoints. (Caution of Hallucination!)

| | *Given prompt*: Paris is known for |
|---|---|
| `https:// medicalxpress. com/` | 150 years of medical excellence, thanks to the collaboration of doctors, nurses, medical students and researchers. It is also known as the home of major medical discoveries such as the operation of the heart, the discovery of insulin, the discovery of the HIV virus, the discovery of the X-ray, the discovery of the... |
| `https: //nytimes.com/` | 4 things: 1. Big. It's the capital of France and also the largest city in Europe. 2. Beautiful. It's a very beautiful city. 3. Rich. It's one of the richest cities in the world. 4. Old. It's one of the oldest cities in the world. |
| `https://reddit. com/` | 4 things, its cuisine, its fashion, its nightlife and its cinema. And I don't mean that only. 😊What if I told you that there is a community of actors who actually act out the scenes of French movies? This community is called REDdit! |

Overall, our results suggest that context-aware pretraining, especially when paired with dynamic guidance strategies at inference, offers a powerful toolset for building more efficient, adaptive, and steerable LLMs. Future work could explore scaling these techniques to larger models and broader metadata sources, as well as integrating contextual signals directly into fine-tuning and instruction tuning pipelines.

**Acknowledgment.** This work was supported as part of the Swiss AI Initiative by a grant from the Swiss National Supercomputing Centre (CSCS) under project ID a06 on Alps. DF would like to thank Diba Hashemi for helpful discussions.

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

# A Limitations and Future Work

In this work, we investigated three different types of metadata. Yet it is not a very thorough investigation. Furthermore, the results we present in the paper are all from 1.5B models on 100B Fineweb-Edu dataset. It remains to be seen if our findings are consistent when scaling up the model sizes and token counts or switching to different data corpora.

Our work points out a lot of interesting findings from observations in our experiments. The mechanics behind the improved performance brought by URL conditioning are still unclear. From our experimental results, we know it is not because of the indicated topic/format from the URLs, or it could be that the LMs have a different understanding of topic and format domains. Future work should investigate the changes in activations between context-free and context-conditioned pretrained models to have a better understanding.

# B Broader Impacts

As discussed in Section 4.2, context-conditioned models are more easily steered. While this capability can be used to guide responses in a desired style or format, it also has the potential to produce harmful or untruthful content. Therefore, these methods should be applied with social responsibility.

# C More Experimental Details

Each of the model training run takes around 800 GPU hours on GH200(120GB).

## C.1 Missing figures and tables

### C.1.1 Acceleration measured by downstream task performances

In addition to Figure 4, we provide the evaluation with respect to the remaining tasks in Figure 7. The acceleration is consistent across different tasks.

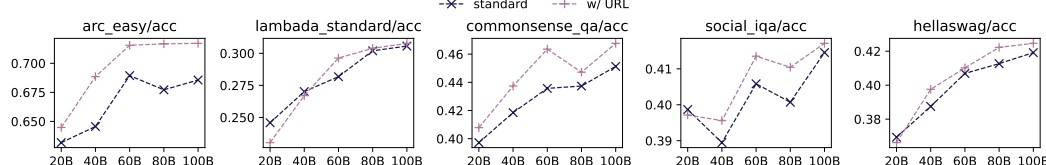

Figure 7: Training acceleration in terms of downstream performances.

### C.1.2 Attention pattern to different parts of contexts

We randomly sample 100 documents from FineWeb-Edu and prepend each with its corresponding URL, quality score, and domain information. These prepared documents are then input to the corresponding context-conditioned pretrained checkpoints; i.e., URL-prepended documents are fed to the URL-conditioned model. We use a batch size of 1 and a sequence length of 4096, without any padding. The attention weights shown in Figure 5 are averaged across all attention heads in the layer and over all non-contextual tokens, and are normalized by the total attention assigned to the contextual portion.

We further visualize the evolution of attention patterns across layers using a heatmap, as shown in Figure 8. Starting from layer 3, we observe a pronounced attention sink on semantically uninformative and commonly shared tokens, such as the URL prefix (`https://`). This pattern persists across other metadata-conditioned checkpoints. We therefore hypothesize that the attention in the first two layers contributes meaningfully to training acceleration.

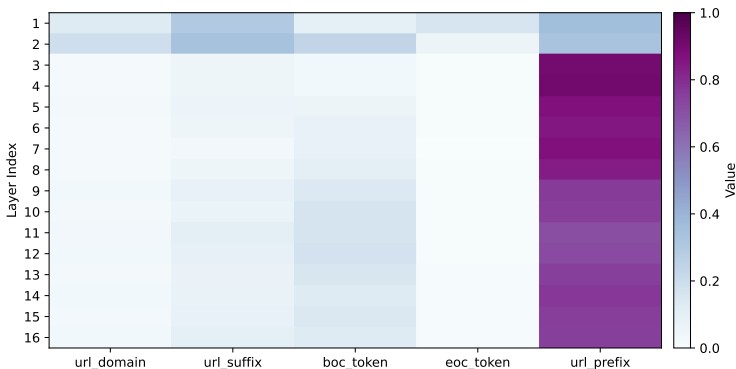

Figure 8: Average attention pattern of URL-conditioned model across layers to different parts of URL context.

Table 8: Example of different continuations given a prompt conditioned on different topics, while the format domain is kept as `Knowledge Article`. The continuations are generated in a context-guided fashion using our pretrained checkpoints. (Caution of Hallucination!)

|  | *Given prompt*: Paris is known for |
| --- | --- |
| Crime & Law | its cultural diversity, and LGBTQ+ people, women, and people of color are often underrepresented. The city is home to many LGBTQ+ neighborhoods, which are underrepresented in many other cities and towns. However, these neighborhoods are particularly vulnerable to violence, abuse, and harassment. |
| Finance & Business | its economic prosperity. In the late 19th and early 20th centuries, Paris was a hotbed of the industrial revolution in Europe, and it continues to be a hub for commerce. The city's robust economy is largely fueled by its high-tech sector, which includes both the manufacturing and technology industries ... |
| Entertainment | its rich musical traditions, with a thriving musical scene that has left an indelible mark on the cultural landscape. From the classical era to the modern era, Paris has been a city of musicians and composers who have shaped the music of the world. |
| Education & Jobs | its prestigious education system. Universities are renowned for their research capabilities, cutting-edge facilities, and excellent faculty. To ensure that universities meet the highest standards of academic excellence, France has implemented various quality assurance mechanisms, including external examinations. These examinations assess the overall quality of a university's academic programs, faculty, and student body. |

### C.1.3   More steering results

We further present the steering results on the DI-conditioned model in Table 8 and Table 9. Given the exact same prompt, the continuation can be steered in markedly different directions depending on the contextual guidance. Topic-based guidance is highly visible, while format-based guidance is more subtle. For example, when steering with "News Article", the output tends to include concrete details about the composer and adopts a more objective tone—different from the more casual and subjective tone of "Personal Blog". Similarly, when guided by "Spam/Ads", the output adopts a promotional style, often highlighting a specific fair in an advertising manner.

### C.2   Prepended context in inference time

Table 9: Example of different continuations given a prompt conditioned on different formats, while the topic domain is fixed as `Entertainment`. The continuations are generated in a context-guided fashion using our pretrained checkpoints. (Caution of Hallucination!)

| | *Given prompt*: Paris is known for |
|---|---|
| News Article | its rich music history, and the city has produced many renowned composers who have left their mark on the world of music. One such composer is Jules Massenet, a French composer who lived from 1842 to 1912. Massenet is considered one of the most important composers of French opera... |
| Personal Blog | its music scene and it is home to some of the most famous musicians in the world. One of the most famous French musicians is Charles Dutoit, who is widely considered to be one of the best French musicians of all time. He has a unique style that blends traditional French music with a modern approach, and he... |
| Spam/Ads | its rich musical traditionits many wonderful fairs, particularly the fairs at the Les Demoiselles d'Avignon, but also at the many other fairs throughout the year. In music, the term "chanson" is often used, especially in the 18th century, to refer to a genre of music that was particularly |

Table 10: Different contexts for Table 3.

| Task | URL context | DI context | QS context |
|---|---|---|---|
| Arc-E/C | `www.factmonster.com/` | Topic: Science & Tech., Format: Knowledge Article | |
| CSQA | `www.factmonster.com/` | Topic: Social Life, Format: Knowledge Article | |
| MMLU | `www.reference.com/` | Topic: Science & Tech., Format: Knowledge Article | Quality Score: 5 |
| PIQA | `www.howstuffworks.com/` | Topic: Social Life, Format: Knowledge Article | |
| SIQA | `www.reddit.com/r/ AskSocialScience/` | Topic: Social Life, Format: Knowledge Article | |
| HS | `www.wikihow.com/` | Topic: Home & Hobbies, Format: Knowledge Article | |
| LBD | `www.wikihow.com/` | Topic: Literature, Format: Creative Writing | |
| WG | `www.reddit.com/r/ explainlikeimfive/` | Topic: Social Life, Format: Personal Blog | |

## C.3   LLM Judge

We use `gpt-4o` from OpenAI API to evaluate continuations given a prompt. The prompt used for scoring medical & health domain continuations is presented in the following block. For the history domain, we modify it correspondingly to suit the topic.

You will be given a prompt and a completion that attempts to continue the prompt. Your task is to judge the quality of the completion based on the following criteria:

Coherence: Does the continuation flow logically and grammatically from the prompt?

Correctness: Is the information medically or scientifically accurate?

Reasonableness: Is the claim plausible and not misleading or exaggerated?

Relevance: Does it directly relate to and complete the idea expressed in the prompt?

Rate the continuation on a scale from 1 to 10, where:

1 = incoherent, irrelevant, or medically incorrect

5 = somewhat relevant or plausible, but lacking clarity or precision

10 = fully coherent, factually accurate, and medically relevant

You should ignore the irrelevant answer if the relevant part makes good continuation. You must not hallucinate or invent facts. Use only information in the prompt and completion.

Respond in this format:

Prompt: question

Continuation: answer

Rating: (your rating, as a float between 1 and 10)

Rationale: (Brief explanation of the score)

