# OpenReview forum: "URLs Help, Topics Guide: Understanding Metadata Utility in LLM Training"
_NeurIPS.cc/2025/Conference — NeurIPS 2025 poster_

### Official Review · Reviewer_3pNV · 2025-06-14

**Clarity:** 3
**Significance:** 2
**Originality:** 2
**Rating:** 3
**Confidence:** 3

**Summary:**

This paper studies pretraining a technique to provide metadata of documents during LLMs. While it is already known that providing URLs can accelerate and enhance pretraining, there has been limited systematic analysis of which types of metadata are truly effective and under what conditions. The authors evaluate three types of metadata: URLs, quality scores predicted by LLMs, and domain information annotated by pretrained classifiers. Their experiments show that only URLs contribute to faster training, whereas the other metadata types can even degrade downstream performance. However, the authors also find that context-aware pretraining can enable more controllable generation under classifier-free guidance.

**Questions:**

* Is there any intuitive explanation for the reason why URL conditioning improves performance on general benchmarks like MMLU?

**Ethical Concerns:**

["NO or VERY MINOR ethics concerns only"]

**Final Justification:**

This paper presents an analysis of LLM pretraining with metadata, such as URLs, and aims to extend the investigation to broader settings. However, due to the following limitations, I consider the paper to be below the threshold for acceptance.

* The paper explores two new types of metadata: quality score and topic/format. However, the motivation for focusing on these particular types is not clearly justified.

* The primary contribution lies in reporting negative results, but the findings are not particularly surprising. Prior work has already shown that metadata is effective only under certain conditions. While this paper extends the analysis to additional settings where metadata is not effective, such findings add limited novelty.

* The new observations have limited practical utility. Although the paper shows that quality score and topic/format metadata can improve the controllability of LLMs, it also shows that these metadata can degrade benchmark performance (Tables 1 and 2). Given the risk of performance degradation, it is unlikely that practitioners would incorporate such metadata in practice.

**Limitations:**

Yes

**Quality:**

2

**Strengths And Weaknesses:**

### Strengths

* This paper investigates context-aware pretraining, an important and broadly applicable technique for enhancing the pretraining of LLMs.
* This paper presents an ablation study evaluating different types of contextual metadata and the conditions under which context-aware pretraining is beneficial. These findings are potentially useful for designing LLM pretraining strategies.
* The paper offers a novel observation that context-aware pretraining can enable more controllable generation when using classifier-free guidance.

### Weaknesses

* Limited diversity in the studied context

There are many types of metadata that can be extracted from websites. However, this paper considers only two types, both annotated by LLMs. The diversity of the selected metadata is limited, and the rationale for focusing on these specific types is not clearly articulated.

* Limited potential impact

The main observations presented in this paper are that (1) only URL metadata improves pretraining performance, and (2) other types of metadata can enhance controllability via classifier-free guidance. However, I am not convinced that these findings are substantially novel or impactful.

While the result that only URL metadata is beneficial for pretraining could be useful for LLM development, the paper does not offer a focused analysis of this negative outcome. In addition, it has already been reported in prior work that metadata conditioning is not universally effective. The contribution would be stronger if this paper included a deeper investigation of this observation across more diverse types of contexts.

Second, although the improved controllability is a novel observation of context-aware pretraining, it is unclear whether this property is a priority for LLM developers when designing pretraining strategies, especially given the potential negative impact on downstream performance. In practice, most developers and users are primarily concerned with overall performance improvements.

* Limited experiments

While the high cost of experimentation is understandable, this paper reports results using only a single, relatively small model architecture with 1.5B parameters, trained with a single set of training configurations. It is unclear how generalizable the findings are to other settings, including larger models or different architectural and training setups.

---

> ### Author Rebuttal · Authors · 2025-07-30
>
> Thank you for your thoughtful feedback. We appreciate the time and effort you invested in reviewing our work. In the following response, we refer to Weakness point i as Wi and Question point i as Qi.
>
> **W1: diversity and rationale of the study context**
>
> We thank the reviewer for their comment and would like to clarify a few points regarding the types of metadata considered in our study.
>
> While it is true that websites can contain a wide range of metadata, these are often not consistently structured or directly accessible for large-scale model training. In our work, we utilized three distinct metadata types: Quality Score and URL domain (both provided by the FineWeb-Edu dataset), and topic/format annotations, which we derived using WebOrganizer.
> Therefore, the statement that “this paper considers only two types, both annotated by LLMs” is not entirely accurate. We investigated three metadata types and explored various combinations to analyze their potential interactions and confounding effects, as detailed in our experiments.
>
> Our focus on these specific metadata types was guided by an initial empirical observation: prepending URLs to documents led to faster pretraining convergence. This raised the question of why URLs might be beneficial.
> Our first hypothesis was that URLs might implicitly signal data quality. This idea was inspired by findings in Physics of Language Models (Section 3.3, [1]), where the authors show that adding a special symbol token to high-quality data can mitigate capacity degradation caused by junk data, thereby enhancing training. To test this, we prepended explicit Quality Scores (on a 1–5 scale), but did not observe a similar acceleration.
>
> We then hypothesized that URLs might encode information about topic and format—for example, the URL domain often reflects the format (e.g., blog, article, dataset), while the URL suffix may hint at the topic. To test this, we annotated our corpus accordingly and ran further experiments. Again, we did not see the same speed-up in training.
> This progression reflects the reasoning behind our choice of metadata and experimental design, and we hope this clarifies our approach.
>
> **W2: potential impact**
>
> We respectfully disagree with the statement that “it has already been reported in prior work that metadata conditioning is not universally effective.” While the study by Higuchi et al. [2], which we also cite in our paper, does raise valid concerns about the effectiveness of metadata conditioning, their conclusions were drawn from a highly controlled setting using synthetic data generated from context-free grammars. In their framework, metadata referred specifically to the generation rule used for each sample, and they found that such metadata was helpful only under limited conditions.
>
> In contrast, our work extends this line of inquiry to **more realistic and diverse forms of metadata**, evaluated at **web-scale** pretraining levels. Given the significant differences in both methodology and experimental scale, we believe the conclusions of Higuchi et al. do not directly generalize to our setting.
>
> Regarding why only URL metadata is beneficial, we refer you to a detailed analysis in our response to **Reviewer RrS9** (**W3** part).
>
> Regarding your comment on whether improved controllability is an important consideration for LLM pretraining strategies, we agree that overall performance remains a central concern for developers and users. However, we argue that improved controllability can lead to enhanced downstream performance, as evidenced by the gains we report in Table 3 with context-conditioned generation.
>
> We believe our work contributes a broader and more practical perspective on **an important axis of data curation**. Beyond quality filtering, leveraging overlooked metadata can further enhance the utility of training data.
>
> **W3: experimental scale**
>
> We acknowledge the concern raised. Due to limitations inherent to our academic setting, it was already a significant challenge to complete the 1.5B parameter experiments. Each experiment on 100B tokens requires over 800 GPU hours. While we recognize that results may not fully generalize to larger-scale models, our computational resources did not permit further scaling. Nonetheless, we believe that the 1B scale remains a reasonable and informative choice. Notably, the predecessor work MeCo[3] conducted the majority of their experiments using a 1.6B Llama model, which supports our design decision. Given that their findings generalized well across models ranging from 600M to 8B parameters, we feel that our choice of scale is a reasonable and pragmatic compromise under the given resource constraints.
>
> Regarding **Q1**—why URL conditioning improves performance on general benchmarks like MMLU—we suspect this is because the model achieves a faster rate of knowledge acquisition and ultimately develops a higher knowledge capacity, as evidenced by the more rapid loss convergence. This acceleration may stem from data contextualization, which reduces the entropy of early token predictions. As shown in Figure 3 of the manuscript, the URL-conditioned model exhibits a steeper loss decline from the very beginning of training.
>
> We hope this explanation clarifies the motivation behind our design choices and the observed results. If you find that our response satisfactorily addresses your concerns, we would be grateful if you could consider raising your score. If any questions remain, we are happy to provide further clarification.
>
>
> [1] Allen-Zhu and Li, Physics of Language Models: Part 3.3, Knowledge Capacity Scaling Laws
>
> [2] Higuchi et al. When Does Metadata Conditioning (NOT) Work for Language Model Pre-Training? A Study with Context-Free Grammars
>
> [3] Gao et al. Metadata Conditioning Accelerates Language Model Pre-training

---

> > ### Comment · Reviewer_3pNV · 2025-08-01
> > **Re: Rebuttal by Authors**
> >
> > Thank you for your response.
> >
> > *  W1: diversity and rationale of the study context
> >
> > I mentioned "this paper considers only two types" by referring to the quality score and topic/format annotation, because URLs have already been studied in prior work.
> >
> > I appreciate the clarification regarding the rationale behind the selection of the two metadata types. From your explanation, I understand that the paper aims to investigate why URLs are beneficial for pretraining. However, if this is the research objective, the construction of the metadata does not align with that goal.
> >
> > Both the quality score and the topic/format annotations incorporate information beyond what is available in the URLs themselves. As such, even if the experimental results were positive, they would not support any direct conclusions about the benefit of URLs. Since the metadata is not derived solely from URLs, the experiments cannot isolate or attribute any observed effects to URL-specific information.
> >
> > If the intent is to analyze the benefit of URLs, a more appropriate approach would be to predict the quality score or topic/format using only the URL, possibly leveraging language models. This would allow for a more targeted analysis of which elements of the URL contribute to performance gains.
> >
> > In summary, I remain unconvinced that the experiments are well designed or well motivated.
> >
> > * W2: potential impact
> >
> > I am aware that the cited work is based on synthetic data and that your work builds upon and extends these studies. However, your explanation does not contradict my original understanding that it has already been reported in prior work that metadata conditioning is not universally effective. If you believe this is incorrect and disagree with the statement, please directly clarify that prior work does not state that metadata conditioning is not universally effective.
> >
> > Regarding the concern about the performance drop, I am referring the results presented in Tables 1 and 2. Table 3 is reasonable for analysis but is not a fair comparison.
> >
> > * W3: experimental scale
> >
> > I acknowledge the limitations imposed by computational resources. I am not placing substantial weight on this limitation in my overall evaluation. However, any claims regarding generalizability should not be made based on your experiments conducted solely with a single small model.
> >
> > * Q1
> >
> > Thank you for the explanation. I understand that this is a challenging question, but it could be interesting to include example responses that illustrate the influence of URLs. However, I do not consider this question important for the overall evaluation and I do not expect the authors to spend much time on it during the discussion period.

---

> > > ### Author Response · Authors · 2025-08-01
> > >
> > > Thank you very much for the quick response! We really appreciate it. Regarding your further comments, we have the following response:
> > >
> > >
> > > **W1: diversity and rationale of the study context**
> > >
> > > The goal of our paper is not to offer a theoretical explanation for why URLs are effective for pretraining, but rather to explore whether other forms of metadata can similarly accelerate training. The rationale we provided was intended to support our experimental design choices; we hypothesized that the two selected metadata types might enhance pretraining efficiency based on the reasoning outlined in our previous response. While these alternatives did not yield the same acceleration benefits as URL conditioning, we believe this negative result is still valuable to the community, as it helps to clarify the limitations and scope of metadata conditioning for pretraining.
> > >
> > >
> > > **W2: potential Impact**
> > >
> > > Regarding the “metadata conditioning is not universally effective”,  prior work **specifically** refers to inference-time limitations—that is, metadata conditioning tends to be beneficial only when longer prompts are available at inference time. In contrast, our study evaluates the universality of metadata conditioning more broadly: across diverse metadata types, training efficiency, and different inference-time settings.
> > > Not all metadata types are effective, both in terms of training speed and downstream performance
> > > URL as metadata is not effective with 0-shot inference, but is effective with 5-shot inference.
> > >
> > > Thank you for the clarification. We acknowledge the performance drop reported in Table 1. However, as shown in Table 2, under 5-shot evaluation—which does not require additional metadata at inference time—both the topic/format-conditioned and quality-score-conditioned models achieve performance comparable to the standard model, while offering the added benefit of enhanced controllability. Notably, 5-shot inference is a widely adopted practice in LLM usage.
> > >
> > > Regarding the practical impact, we recommend URL-prepending as a pretraining strategy for LLM developers. Unlike the two-phase approach in MeCo (URL-conditioned pretraining followed by URL-free “cooldown” training), we demonstrate that a 90/10 mixture of URL-conditioned and URL-free data during training is equally effective. This approach enables URL-free inference at any checkpoint, eliminating the need for a separate cooldown phase such as the 10B-token URL-free training stage used previously.
> > >
> > >
> > > **W3: experimental scale**
> > >
> > > We thank you for the further explanation of W3. We have not overclaimed our findings. In Appendix A Limitations and Future Work, we stated explicitly that “It remains to be seen if our findings are consistent when scaling up the model sizes and token counts or switching to different data corpora” (Line 462-463).
> > >
> > > We acknowledge that this limitation could be made more prominent, and we will ensure it is emphasized more clearly in the updated manuscript.
> > >
> > > Thank you again for your time and input! If any questions remain, we are happy to provide further clarification or have more discussions.

---

> > > > ### Comment · Reviewer_3pNV · 2025-08-02
> > > > **Re: Official Comment by Authors**
> > > >
> > > > Thank you for the response. I still have two main concerns, both of which stem from the selection of the target metadata: quality score and topic/format. While I understand the authors' explanation that these metadata types are motivated by information present in URLs, I remain unconvinced that focusing on these two specific forms of metadata offers a significant contribution to advancing the understanding of pretraining with metadata.
> > > >
> > > > Given these concerns, I am inclined to keep my original score.
> > > >
> > > > * W1: diversity and rationale of the study context
> > > >
> > > > > The goal of our paper is not to offer a theoretical explanation for why URLs are effective for pretraining, but rather to explore whether other forms of metadata can similarly accelerate training
> > > >
> > > > Thank you for the clarification. Yes, this aligns with my initial understanding, and I considered that exploring these two types of metadata is not sufficiently comprehensive, particularly given that this is a negative result paper. While negative results can be valuable contributions, they typically require more extensive and comprehensive experimentation to support their significance.
> > > >
> > > > For example, if you had conducted an exhaustive analysis of information that URLs may encode and demonstrated negative results across all of them, that could have been a surprising and valuable finding. Such a result might motivate further investigation into the role of URLs in pretraining. However, since URLs can contain a wide range of information, the rationale for focusing exclusively on just two types of metadata is still not sufficiently justified.
> > > >
> > > > In addition, as I mentioned in my previous response, the new metadata is not extracted from URLs, and therefore, the experiments do not support any direct claims about the role of URLs in pretraining. This further weakens the contribution of the negative results presented in the paper.
> > > >
> > > > * W2: potential Impact
> > > >
> > > > > Regarding the “metadata conditioning is not universally effective”, prior work specifically refers to inference-time limitations
> > > >
> > > > Thank you for the clarification. I acknowledge that this paper presents this partially known phenomenon in a broader range of settings. However, I do not find the results to be sufficiently novel or surprising.
> > > >
> > > > This is closely related to the first point. Although the setting is diverse compared to prior work focusing on URLs, I still do not see a clear justification for studying the quality score and topic/format, while I understand your explanation about the motivation coming from URLs. There is no strong reason to expect that the two new types of metadata studied would be effective, and so the negative results are less surprising.
> > > >
> > > > > However, as shown in Table 2, under 5-shot evaluation—which does not require additional metadata at inference time—both the topic/format-conditioned and quality-score-conditioned models achieve performance comparable to the standard model
> > > >
> > > > I believe that the significant performance drop from 47.8 with "+ <boc> URL <eoc>" to 46.0 with "+ <boc> URL, QS, DI <eoc>" should be taken more seriously than the optimistic interpretation based on the observation that "+ <boc> QS <eoc>" and "+ <boc> DI <eoc>" individually yield performance comparable to the original.

---

> > > > > ### Author Response · Authors · 2025-08-02
> > > > >
> > > > > Thank you for the thoughtful and engaging discussion.
> > > > >
> > > > > - Scope of Metadata
> > > > >
> > > > > We recognize that our metadata selection is necessarily limited. High-quality, structured metadata is scarce at web scale, so we concentrated on the few categories that are consistently available.
> > > > > If you can point to additional metadata types that are both impactful and practical to obtain, we would welcome your suggestions and will explore them in future work.
> > > > >
> > > > > - Rationale for the Non-URL Experiments
> > > > >
> > > > > The two non-URL runs were not designed to explain URL effectiveness. As for the rationale that we presented in the last response, we hope to keep the experimental design fully transparent, as requested. "Since URLs can contain a wide range of information, the rationale for focusing exclusively on just two types of metadata is still not sufficiently justified." We would like to hear more insights on this comment.
> > > > >
> > > > > A detailed analysis of why URLs, specifically, speed up pre-training is provided in our reply to **Reviewer RrS9 (section W3)**. In brief, the very limited attention allocated to salient portions of the QS and DI metadata across all layers likely accounts for their weaker results.
> > > > >
> > > > > - Interactions Between Multiple Metadata Signals
> > > > >
> > > > > With more than one metadata type, there may be a conflicting signal contained in the metadata, thus resulting in a performance drop. We are aware of this effect. Thus, we recommend URL-conditioned pretraining for LLM practitioners. We also outline additional guidance mechanisms that could help models exploit the rich information embedded in URLs or other metadata types more effectively.
> > > > >
> > > > > We hope these clarifications address your concerns and look forward to any further feedback.

---

> > > > > > ### Comment · Reviewer_3pNV · 2025-08-03
> > > > > > **Re: Official Comment by Authors**
> > > > > >
> > > > > > Thank you for your response. However, I remain unconvinced that the experimental design of the paper is sufficiently well-motivated or well-justified.
> > > > > >
> > > > > > Given the observation that URLs are the only known type of metadata shown to improve performance, I believe the most natural and important research question is to determine which specific information within URLs contributes to this improvement. I encourage future work to focus on analyzing the content of URLs to identify the key factors responsible for the observed performance gains. Specifically, I expect such analyses to extract information directly from URLs in order to assess which components are effective. Once a particular type of information is identified as beneficial, it would then be meaningful to explore ways to enhance or generalize it using additional sources of metadata, such as those studied in this paper.
> > > > > >
> > > > > > Although negative results can be a valuable contribution, I still consider that the choice of the two types of metadata studied in this paper is not sufficiently justified. As a result, the findings do not significantly improve our understanding of training LLMs with metadata. Negative results can represent a significant contribution if there is a strong justification for expecting that the studied metadata would improve performance, or if the metadata is analyzed in a sufficiently comprehensive manner. In my view, neither of these conditions is met in this paper.

---

### Official Review · Reviewer_oR6w · 2025-07-03

**Clarity:** 3
**Significance:** 4
**Originality:** 3
**Rating:** 5
**Confidence:** 3

**Summary:**

Recent works have shown that adding metadata to pre-training documents can be beneficial. However, it is unclear what kind of metadata is beneficial, in what way it is beneficial, and how this depends on how we evaluate/use the trained model. This paper studies the role of metadata in more depth. They find that adding URL context speeds up training, while other metadata (quality as well as topics/formats) does not have any effect. In terms of downstream performance, URL conditioning helps more on 5-shot than 0-shot evaluation. They also find that all types of metadata are able to help more effectively steer generations---for instance, asking a question about Julius Caesar conditioned on fashion and beauty blogs results in an answer that involves makeup.

**Questions:**

1. Is there a training acceleration figure for MeCo?
2. When coming up with the metadata for a test sample at inference time, is there a scalable procedure?
3. What happens when we do context-aware generation with URLs/topics/formats that are not in the training data?

**Ethical Concerns:**

["NO or VERY MINOR ethics concerns only"]

**Limitations:**

Yes

**Quality:**

3

**Strengths And Weaknesses:**

**Strengths**

Quality:
- The paper presents a comprehensive study on how to incorporate metadata into pretraining, and how to utilize models that are trained in this way.
- The finding that metadata can steer generations is quite interesting and could have implications for personalization.

Significance:
- This sort of systematic evaluation is valuable, as the behavior of metadata conditioning is not well-understood.

**Weaknesses**

Quality:
- Regarding the results comparing 5-shot vs 0-shot performance, I am not sure if the conclusion is that URL conditioning only makes a difference with longer prompts, or if there is some other effect of ICL going on.
- Is there a training acceleration figure for MeCo? (as mentioned in line 200) it would be good to see this to validate MeCo.
- When coming up with the metadata for a test sample at inference time, is there a scalable procedure or does it involve human annotation? For instance, suppose a user has a very out-of-distribution query; what metadata would we use?
- What happens when we do context-aware generation with URLs/topics/formats that are not in FineWeb? I.e., is the context-conditioning based on memorization in the pre-training data, or some understanding of what specific URLs/topics/formats "mean"?

Clarity:
- Some results in section 4 do not reference their corresponding tables/figures.

Originality:
- The work does not exactly propose a novel approach, but this drawback is outweighed by the systematic analysis provided.

---

> ### Author Rebuttal · Authors · 2025-07-30
>
> Thank you for your detailed and encouraging feedback. We appreciate the time and thought you dedicated to reviewing our work. In our response below, we refer to Weakness point i as Wi and Question point i as Qi.
>
> **W1: Is enhanced downstream performance merely due to ICL?**
>
> We do not believe the gains are solely attributable to ICL. As shown in Table 2, only the URL-conditioned model demonstrates a significant performance improvement with longer prompts. If ICL alone were responsible, we would expect similar benefits across all models. This suggests that URL conditioning plays a distinctive role, especially with extended context.
>
>
> **W2 & Q1: Training curve of MeCo.**
>
> Yes, we have recorded the MeCo training curve using Weights & Biases. As suggested, we will include this curve in the updated version of the paper.
>
> **W3 & Q2: Scalable metadata generation at inference time**
>
> In our current implementation, metadata at inference time was produced through human annotation. In some cases, we also leveraged ChatGPT for assistance. Our used metadata at inference time is recorded in Table 10.  For scalability, we believe that automated approaches—such as using API calls with in-context examples—can replicate this process effectively.
>
> Notably, the prepended URL does not have to be real. For highly out-of-distribution (OOD) queries, a fabricated yet plausible URL can suffice. Quality Score can be set to the highest value by default, and the quality/format domain can be assigned using the WebOrganizer tool, which operates at web scale and offers sufficient coverage.
>
> **W4 & Q3: OOD metadata?**
>
> As mentioned in our response to W3, the metadata—such as URLs—does not need to exist in the pretraining data. Fabricated URLs can still guide generation effectively, which suggests that the model is leveraging latent semantic understanding rather than simply relying on memorization.
>
> We include examples involving OOD topics and URLs in the Tables below (noting the potential for hallucinations).
>
> When given novel topics such as fruits or animals, the model appropriately centers the continuation around the topic. However, limitations remain: for instance, when prompted with Dark Humor, the model fails to grasp the topic meaningfully and defaults to generic content (e.g., a standard description of Paris).
>
> Similarly, when we test with fabricated URLs, the URL-conditioned model still demonstrates a good grasp of their intended semantic meaning.
>
>
> Table 1: OOD topic guidance. The prompt is “Paris is known for”.
> | Topic | Continuation|
> | --- | --- |
> |Fruits | its delicious and diverse culinary offerings, and one fruit that’s particularly special is the pear. Native to the Mediterranean and Middle East, pears have a rich history and are a popular ingredient in many dishes. In this article, we’ll explore the many reasons why pears are so special…|
> | Animals | its historical and architectural monuments, as well as for its rich gastronomy. It is also one of the countries with the highest concentration of zoos in the world. The Paris Zoo is considered one of the most important zoos in the world, located in the heart of the city in the Parc des Buttes-Cha"...|
> |Dark Humor | its numerous historical monuments and landmarks. One such landmark is the Basilica of Notre Dame. This cathedral is located in the heart of Paris and is one of the most famous landmarks in the city. |
>
> Table 2: OOD URL guidance. The prompt is “Paris is known for”.
> | Url | Continuation|
> | --- | --- |
> | https://easy-science-for-kids.com/ |  the discovery of helium which is found in the atmosphere. Helium was known to be a unique element at that time. Helium is known to be a very important element for humans and other living beings…|
> | https://www.modern-fashion-tips.com | fashion, and for a good reason, fashion is a field that involves designing, manufacturing and selling of clothing and accessories. Fashion design has a lot of components, such as the concept, pattern, fabric, color, pattern, size, and fit. Fashion design can also be characterized by the trends, styles, and …|
>
>
> We will make sure that all the Tables/Figures are referenced in Section 4.
>
> We hope this clarifies our decisions and findings. If any outstanding issues remain, please let us know, and we'll be happy to provide further clarification.

---

> > ### Comment · Reviewer_oR6w · 2025-08-06
> >
> > Thank you for your response. I have no additional questions and will maintain my score.

---

### Official Review · Reviewer_RrS9 · 2025-07-20

**Clarity:** 3
**Significance:** 3
**Originality:** 3
**Rating:** 4
**Confidence:** 3

**Summary:**

This paper investigates whether incorporating metadata, such as URLs, quality scores, and domain info, during LLM pretraining improves training efficiency and downstream performance.
The paper conducts a systematic study using context-conditioned pretraining, where metadata is prepended but excluded from the loss function.
They find that only URLs help accelerate training and improve performance, particularly for longer prompts during inference.
Domain metadata, while not helpful in training speed or general performance, provides strong controllability during inference.

**Questions:**

- Can the findings generalize to larger models?

**Ethical Concerns:**

["NO or VERY MINOR ethics concerns only"]

**Final Justification:**

The authors' response has addressed some of my concerns.

**Limitations:**

Yes.

**Quality:**

3

**Strengths And Weaknesses:**

## strength

- The paper introduces a novel experiment setup by evaluating different metadata in a controlled pretraining framework.
- The paper is well-organized with clear motivation and detailed descriptions of methods and experiments.
- The paper conducts experiments and gives empirical findings.

## weakness

- The central issue is that all experiments in the paper now use only 1.5B language models (Llama). Results may not generalize to larger models. This significantly hampers the contribution.
- The experiments only consider a single dataset (FineWeb-Edu).  Results may not generalize to more diverse corpora.
- Why URLs help more is not deeply analyzed. This may weaken the impact of the findings.

---

> ### Author Rebuttal · Authors · 2025-07-30
>
> Thank you for your thoughtful feedback. We appreciate the time and effort you invested in reviewing our work. In the following response, we refer to Weakness point i as Wi and Question point i as Qi.
>
> Regarding **W1**:
>
> We acknowledge the concern raised. Due to limitations inherent to our academic setting, it was already a significant challenge to complete the 1.5B parameter experiments. Each experiment on 100B tokens requires over 800 GPU hours. While we recognize that results may not fully generalize to larger-scale models, our computational resources did not permit further scaling. Nonetheless, we believe that the 1B scale remains a reasonable and informative choice. Notably, the predecessor work MeCo[1] conducted the majority of their experiments using a 1.6B model, which supports our design decision. Given that their findings generalized well across models ranging from 600M to 8B parameters, we feel that our choice of scale is a reasonable and pragmatic compromise under the given resource constraints.
>
> Regarding **W2**:
>
> We contend that FineWeb-Edu is already a sufficiently diverse corpus. It is derived from Common Crawl, first processed into FineWeb (a cleaned and deduplicated version), and then further filtered for educational quality to yield FineWeb-Edu [2]. This filtering makes FineWeb-Edu more data-efficient than FineWeb, allowing us to achieve comparable downstream performance using fewer tokens.
>
> Regarding **W3**:
>
> We were similarly intrigued by the observation that URL was the only metadata type that consistently yielded a performance benefit. Since the submission, we have begun investigating this phenomenon and have formulated two working hypotheses:
>
> - H1: Prepending URLs significantly alters the model’s attention distribution.
> - H2: Prepending URLs enhances the structure of the learned latent representations, leading to more distinguishable and fine-grained embedding spaces.
>
> To evaluate H1, we performed both visual and quantitative analyses of the attention patterns.
>
> For clarity, consider the example URL (note this is not an external link, just an example):
>
>  "https://en.wikipedia.org/wiki/Metadata".
>
> Here, "https://" is the prefix, "en.wikipedia.org" is the domain, and "wiki/Metadata" is the suffix. Often, URL domain carries format information and URL suffix carries topic information.
>
> Surprisingly, across all three models, we observe relatively little attention given to the semantically meaningful components of the metadata. More specifically, attention patterns in the URL-conditioned model vary by layer (see Table 1):
> - In layers 1–2, the model assigns higher attention to the URL domain and URL suffix, which carry informative content.
> - From layer 3 onward, however, attention shifts predominantly to common URL prefixes like “https://”, which are shared across most examples and carry little semantic value.
>
> In contrast, for models conditioned on Quality Score and Topic/Format, the attention allocated to informative metadata remains _consistently_ low across all layers (see Tables 2 and 3).
>
> We hypothesize that the early-layer attention to meaningful URL components may play a key role in the effectiveness of URL prepending, even if later layers focus on less informative tokens.
>
> Table 1: Attention weight distribution for URL-conditioned model.  Context is “\<boc\>\<URL\>\<eoc\>”. Non-context is the text following the context.
>
> || non-context | context |  \<boc\> & \<eoc\> | URL prefix | URL domain | URL suffix |
> | --- | --- |--- |--- |--- |--- |--- |
> |1st layer | 79.3 | 18.9 | 5.2| 4.8 | 3.5 | 5.0 |
> |middle layer| 55.8 | 43.9 | 4.7| 37.4| 0.6 | 1.1|
> |last layer| 67.5| 32.2 |4.6| 24.5 | 0.9 |2.0 |
>
> Table 2: Attention weight distribution for Quality Score-conditioned model. Context is “\<boc\>Quality Score: \<QUALITY_SCORE\>\<eoc\>”
> || non-context | context | \<boc\> & \<eoc\> | \<QUALITY_SCORE\> |
> | --- | --- |--- |--- |--- |
> |1st layer | 85.8 | 12.3 | 10.4 | 0.2 |
> |middle layer| 39.6 |55.1| 23.8| 0.3|
> |last layer| 59.3| 27.3 |13.0 | 0.1|
>
> Table 3: Attention weight distribution for Topic/Format-conditioned model. Context is “\<boc\>Topic: \<TOPIC\>, Format: \<FORMAT\>\<eoc\>”
>
> || non-context | context | \<boc\> & \<eoc\> | \<TOPIC\> | \<FORMAT\> |
> | --- | --- |--- |--- |--- |--- |
> |1st layer | 83.3 | 15.1 | 5.1 | 1.0 | 1.4 |
> |middle layer| 54.0 |42.7|2.0 | 0.5 | 0.2|
> |last layer| 66.9 |23.0 | 6.1|0.4 |0.3|
>
> To evaluate H2, we sampled 100 documents from each of 100 distinct URL domains within the FineWeb-Edu dataset, yielding a total of 10,000 documents. We did a 90-10 train/test split. We then trained linear probes on the hidden activations from each layer to classify documents by their originating domain.
>
> While we observed a general upward trend in probing accuracy with increasing layer depth, there was no significant difference between the URL-conditioned model and the standard pretrained model (Table 4). This suggests that, despite URL prepending, the structure of latent representations—at least as measured by domain separability—does not differ markedly between the two models.
>
> Table 4: Test probing accuracy across layers.
> |Layer index| URL-conditioned | Standard |
> | --- | --- |--- |
> |1st layer | 32.97 | 32.48 |
> |middle layer | 61.99 | 62.49 |
> |last layer | 64.18 | 65.04 |
>
> Our preliminary exploration suggests that the altered attention patterns may contribute to the performance gains observed with URL prepending. However, further investigation is needed to draw a more comprehensive and conclusive understanding of this effect. We hope this investigation provides some insights!
>
> We hope this clarifies our decisions and findings. If you feel that our response sufficiently addresses your concerns, we kindly request that you consider raising your score. If any outstanding issues remain, please let us know, and we'll be happy to provide further clarification.
>
>
> [1] Gao et al. Metadata Conditioning Accelerates Language Model Pre-training
>
> [2] Penedo et al. The FineWeb Datasets: Decanting the Web for the Finest Text Data at Scale

---

> > ### Comment · Reviewer_RrS9 · 2025-08-07
> >
> > Thank you for your response. It has addressed some of my concerns, and I will raise my score.

---

### Decision · Program_Chairs · 2025-09-17

**Decision:**

Accept (poster)

**Comment:**

This paper investigates the role of incorporating metadata during LLM pretraining, focusing on URLs, quality scores, and topic/format information. The authors systematically evaluate the effects of these metadata types by prepending them during training but excluding them from the loss function. Their results show that only URLs accelerate training and improve downstream performance, particularly with longer prompts, while topic/format and quality scores mainly enhance controllability of generations rather than efficiency or accuracy.

In general, the paper is well-motivated and clearly written, with careful experimental design and a systematic analysis of different metadata types. The empirical results provide useful insights into when and how metadata can contribute to training efficiency and controllable generation. However, as pointed out by the reviewers, there are some concerns about the limited experimental scale (restricted to 1.5B models and one dataset), the relatively narrow scope of metadata types studied, and the lack of deeper analysis on why URLs are uniquely effective. Despite these limitations, the paper makes a valuable empirical contribution to the understanding of metadata conditioning in LLM pretraining. I therefore recommend accepting the paper.